# Different RNA Elements Control Viral Protein Synthesis in Polerovirus Isolates Evolved in Separate Geographical Regions

**DOI:** 10.3390/ijms232012503

**Published:** 2022-10-19

**Authors:** Manuel Miras, Miguel A. Aranda, Verónica Truniger

**Affiliations:** 1Centro de Edafología y Biología Aplicada del Segura, Consejo Superior de Investigaciones Científicas (CEBAS-CSIC), 30100 Murcia, Spain; 2Department of Molecular Physiology, Heinrich Heine University of Düsseldorf, 40225 Düsseldorf, Germany

**Keywords:** cap-independent translation, polerovirus, CABYV, 3′-CITE, RNA structure, RNA element, plant virus, translation initiation, translational enhancer

## Abstract

Most plant viruses lack the 5′-cap and 3′-poly(A) structures, which are common in their host mRNAs, and are crucial for translation initiation. Thus, alternative translation initiation mechanisms were identified for viral mRNAs, one of these being controlled by an RNA element in their 3′-ends that is able to enhance mRNA cap-independent translation (3′-CITE). The 3′-CITEs are modular and transferable RNA elements. In the case of poleroviruses, the mechanism of translation initiation of their RNAs in the host cell is still unclear; thus, it was studied for one of its members, cucurbit aphid-borne yellows virus (CABYV). We determined that efficient CABYV RNA translation requires the presence of a 3′-CITE in its 3′-UTR. We showed that this 3′-CITE requires the presence of the 5′-UTR *in cis* for its eIF4E-independent activity. Efficient virus multiplication depended on 3′-CITE activity. In CABYV isolates belonging to the three phylogenetic groups identified so far, the 3′-CITEs differ, and recombination prediction analyses suggest that these 3′-CITEs have been acquired through recombination with an unknown donor. Since these isolates have evolved in different geographical regions, this may suggest that their respective 3′-CITEs are possibly better adapted to each region. We propose that translation of other polerovirus genomes may also be 3′-CITE-dependent.

## 1. Introduction

Cucurbit aphid-borne yellows virus (CABYV) was first described in the early 1990s in France [1] and later identified as one of the most common viruses found in open field cucurbit crops in many other countries, such as Spain, Iran, Greece, Morocco, Egypt, Tunisia, Taiwan, Korea, and China [1,2,3,4,5,6,7,8]. Recently, it has also been reported in Papua New Guinea [9], Brazil [10], Germany [11], and Indonesia [12]. CABYV is one of the prevalent viruses in cucurbit crops, and is often found in mixed infections [2,13]. Its main host range includes cucurbits, such as melon, cucumber, squash, pumpkin, and watermelon, but it also infects agronomically important non-cucurbit species, such as lettuce (*Lactuca sativa*) and fodder beet (*Beta vulgaris*), and several weeds [1]. This virus is phloem-limited, cannot be mechanically inoculated, and is transmitted in nature in a persistent, non-propagative manner by the aphids *Aphis gossypii* and *Myzus persicae* [1]. Very recently, a new recombinant isolate of CABYV from Brazil has been shown to be transmitted by whiteflies [14]. Plants infected with CABYV show symptoms, such as yellowing and thickening of basal and older leaves, and often flower abortion. CABYV is a member of the polerovirus genus, which belonged together with the luteovirus and enamovirus genera, to the *Luteoviridae* family that was recently abolished. While the luteovirus genus was reassigned to the *Tombusviridae* family, the polero- and enamovirus genera were reassigned to the *Solemoviridae* family [15]. The CABYV genome is a single-stranded positive-sensed RNA molecule of 5.7 kb comprising six open reading frames (ORF). The 5′-proximal ORFs resulting in proteins P0, P1, and the ribosomal frameshift protein P1–P2, are translated from genomic RNA. A subgenomic RNA serves as the template for translation of the 3′-proximal ORFs, resulting in proteins P3, P4 and the readthrough protein P3–P5. Possible protein functions, deduced from amino acid sequence similarity with other polerovirus proteins, are PO-post-transcriptional gene silencing suppressor and pathogenicity enhancer, P1–regions of similarity with serine protease and viral genome-linked protein (VPg), P1-P2-RNA-dependent RNA polymerase (RdRP), P3-coat protein (CP), P4-movement protein (MP), P3-P5-role in transmission by aphids [16,17,18]. Recently, an additional ORF adjacent to ORF3, ORF3a, was discovered in the polerovirus turnip yellows virus. It was suggested to play a role in intercellular movement and to be generally expressed in poleroviruses [19]. The CABYV genome, such as that of other polerovirus genomes, has no poly(A)-tail and supposedly no 5′-cap [20,21]. Phylogenetic analyses of CABYV isolate sequences have allowed their classification into the Asian, Mediterranean, and Brazilian groups (Costa et al., 2019; Kassem et al., 2013; Maachi et al., 2022) [10,22,23]. Luteoviruses are recombination prone, thus in the case of CABYV, the continuous generation of recombinant strains has led to its great variability (Costa et al., 2019; Kassem et al., 2013; Kwak et al., 2018; Shang et al., 2009; Xiang et al., 2008) [6,10,22,24,25].

Although most eukaryotic mRNAs depend on the 5′-cap structure and the 3′-poly(A) tail for their translation, only about 20% of known positive-strand RNA viruses have genomic and subgenomic RNAs including these features [26]. Thus, viral mRNAs have alternative mechanisms to recruit the translational machinery of the host for its own translation initiation, allowing them to compete with host mRNAs and to avoid defense mechanisms that act at the translation level. Some of them use their 5′- and/or 3′-termini in alternative gene expression strategies [27,28,29]. Plant viruses from the *Tombusviridae* family lack both the cap and poly(A) tail, and several of its species have been shown to harbor modular functional RNA elements in their 3′-end able to control their cap-independent translation, called cap-independent translational enhancer elements or 3′-CITEs [29,30]. The general mechanistic steps of 3′-CITEs involve recruitment of the eukaryotic translation initiation factors (eIFs) at the 3′-CITE, and their delivery near the translation start site [31,32]. To date, seven different classes of 3′-CITEs have been described. They differ in sequence, structure, and translation enhancing mechanisms. The last class of 3′-CITE described was identified in a melon necrotic spot virus (MNSV, *Tombusviridae*) isolate able to overcome the eIF4E-mediated resistance in melon, MNSV-N [33]. MNSV proteins have been shown to be synthesized by cap-independent eIF4E-dependent translation controlled by a 3′-CITE present in its genomic 3′-UTR [34,35]. The resistance-breaking capacity of the new isolate MNSV-N was shown to depend on a sequence insertion in its 3′-UTR that had the capacity to act as a 3′-CITE in resistant melon. This sequence was highly similar to the beginning of the 3′-UTR of CABYV, and strong evidence was found that it had been acquired by MNSV through interfamilial recombination with part of the 3′-UTR from an Asian CABYV isolate [33]. This important study showed that 3′-CITEs are modular transferrable RNA elements and that its acquisition can associate with a selective advantage for the new recombinant virus.

Since the RNA sequence from CABYV acquired by MNSV-N was shown to be a functional RNA element able to enhance the cap-independent translation of this viral genome, we hypothesized that it would also act as a 3′-CITE in its donor. Thus, in this manuscript we show that the CABYV 3′-UTR is able to enhance the cap-independent translation of a reporter gene, and identify different 3′-CITEs, depending on the group to which the CABYV isolate belongs. We show that these 3′-CITEs need the presence of the 5′-UTR *in cis* but do not need eIF4E for their activity, and that they are required for efficient virus multiplication.

## 2. Results

### 2.1. Conservation of the 3′-UTR Sequences of CABYV Isolates 

Following phylogenetic analyses of the complete genome of CABYV isolates, these were classified into two groups, Asian and Mediterranean [24]. We performed a phylogenetic analysis of the 3′-UTRs of CABYV sequences available in Genbank (Figure 1A) and observed that the separation into these two groups was maintained. After a more precise analysis of the 3′-UTR sequences following its alignment, it was observed that while, in the first half of the 3′-UTR, the sequences varied considerably (until nucleotide 86 (Mediterranean) and 89 (Asian)), in the second half, these were highly similar between both groups (Figure 1B). To simplify this figure, only 8 sequences corresponding to the Asian group, but all of the ones available in Genbank for the Mediterranean group, are shown. For each group separately, the 3′-UTR sequences were highly conserved (Appendix A). Interestingly, the sequence inserted into the 3′-UTR of MNSV-N, shown to have 3′-CITE activity, was located in the variable region of Asian CABYV, as it comprised the first 60 nt of its 3′-UTR [33]. 

These previous results, together with the alignments shown here, suggested that the variable first nucleotides of the CABYV 3′-UTRs (60–90 nts) of Asian and Mediterranean isolates could harbor different 3′-CITEs. 

### 2.2. Identification of 3′-CITEs in CABYV 3′-UTRs

To study the possible role of the CABYV 3′-UTRs in cap-independent translation, the firefly luciferase (luc) gene was flanked by the 5′- and/or 3′-UTRs of CABYV. The 3′-UTRs of the genome of Spanish and French CABYV isolates are identical, representing the Mediterranean isolates (CABYV-Sp) here, while the 3′-UTRs of Asian CABYV isolates are highly conserved (Appendix A), with CABYV Xinjiang (CABYV-Xin) representing these in our experiments. The 20 nt long 5′-UTR genomic sequence of the Spanish and Xinjiang isolates are identical, and it is nearly invariant in all CABYV isolates (Appendix A). The cap-independent translation efficiency of the in vitro transcribed uncapped luc-RNAs was assayed in vivo in melon protoplasts. The luciferase activities measured corresponded to the translation efficiency and are represented as vertical columns in Figure 2 relative to the 5′-luc-3′-UTR construct of CABYV-Spain. Transcripts with only one of the UTRs flanking the luc-gene showed low translation efficiency (between 18% and 6% for the 5′- and 3′-UTRs, respectively, second to fourth column). By contrast, the presence of both CABYV UTRs increased the translation efficiency of these uncapped RNAs, nearly 6-fold for CABYV-Sp (fifth) and 9-fold for CABYV-Xin (eight column). This result suggested that both 3′-UTRs contained a 3′-CITE able to control cap-independent translation in vivo. Its translation efficiency was 2.5–3.5-times lower than that of a capped construct.

Deletion of the first 60 nt of the 3′-UTRs in these 5′-luc-3′-UTR constructs reduced the translation efficiency to a similar level as that obtained in the absence of the complete 3′-UTR (sixth and ninth columns). On the other hand, RNAs with only the first 60 nt of the 3′-UTRs added to the 3′-end of the 5′-UTR-luc constructs showed similar translation efficiencies as the RNAs with both complete 3′-UTRs (seventh and tenth columns). These results allowed us to conclude that these 60 nt were sufficient for cap-independent translation control. Thus, these results confirm that the 3′-CITE identified as active RNA element in MNSV-N (named CXTE, X for Xinjiang) is also working as a 3′-CITE in Asian CABYV isolates, while Mediterranean isolates also have a 3′-CITE that differs in sequence, now named CMTE (M for Mediterranean). Additionally, Figure 2 shows that the CXTE is more efficient in enhancing translation than the CMTE. The translation efficiencies determined for constructs with chimeric 3′-UTRs (where the 3′-CITEs were interchanged, CXTE for CMTE and *vice versa*, last two columns in Figure 2) allow us to first conclude that these 3′-CITEs can be interchanged, and second, that the efficiency in translation enhancement depends on the 3′-CITE itself. 

Similar to melon protoplasts, also in cucumber and *Nicotiana benthamiana* protoplasts, the translation efficiency of the 5′-luc-3′-UTR construct including the CXTE, was higher than that including CMTE (Appendix A; 2.2- and 1.8-fold, respectively). On the contrary, in *Arabidopsis thaliana* Col1 protoplasts, the translation efficiency of the construct including the CXTE was lower than that including the CMTE (Appendix A; 0.5-fold), suggesting that 3′-CITE activities vary in cells of different hosts. 

Several authors have hypothesized, and support has been provided by the first direct proof provided with our studies on MNSV-N, that 3′-CITEs are modular transferrable RNA elements in nature [33,39]. Thus, we analyzed the possibility that the different CABYV 3′-CITEs were interchanged through recombination events, with a recombination occurrence analysis using RDP4 software, which implements several recombination-detecting algorithms [40]. As shown in Figure 3, a recombination event with an unknown donor, including the first part of the 3′-UTR with very high statistical significance (*p* < 0.00001), was detected by RDP4, strongly suggesting that this first part of the 3′-UTR of Mediterranean CABYV, including the 3′-CITE, was acquired through recombination. A nucleotide BLAST search with the recombined new sequence did not result in any matches apart from Mediterranean CABYV. 

### 2.3. Translation Mediated by the CABYV 3′-CITEs Is eIF4E-Independent

The CXTE in MNSV-N was shown to be active in the absence of eIF4E [33], giving this isolate the capacity to infect resistant melon, a resistance that is mediated by a single amino acid change in eIF4E [41]. Here, we studied if the CABYV 3′-CITEs was dependent on eIF4E for their translation enhancement activities, by analyzing the translation efficiencies of the different luc-constructs, first in protoplasts of resistant melon. The translation efficiencies of the luc constructs harboring a CABYV 3′-CITE were similar in protoplasts from resistant melon (dark grey) as from susceptible melon (light grey columns) (Figure 4A), suggesting that these 3′-CITEs could function either with the two different eIF4E variants expressed by susceptible or resistant melon, or independently of eIF4E. This was further studied by analyzing their activities in melon protoplasts of an eIF4E knock-down line expressing a hairpin construct that targeted, and thus silenced, melon eIF4E; eIF4E expression had been previously shown to be reduced more than six-fold in these plants [42] (Rodríguez-Hernández et al., 2012). Previously, we had shown that translation of the MNSV genome was controlled by 3′-CITEs, being eIF4E-dependent for isolate MNSV-Mα5 [34], and independent of eIF4E for resistance-breaking isolates MNSV-N and MNSV-264 [33,42,43]. Thus, the construct of the luc-gene, flanked by 5′- and 3′-UTRs of MNSV-Mα5, was used as eIF4E-dependent translation control, and the constructs with the UTRs of MNSV-N and MNSV-264 as eIF4E-independent translation controls. As observed in Figure 4B, low eIF4E-expression (dark grey columns) associates with the reduced translation efficiency of only the MNSV-Mα5 construct when compared to translation under wild-type eIF4E expression (light grey columns). Thus, cap-independent translation controlled by either CABYV 3′-CITEs were not affected by the reduced eIF4E expression, suggesting that their activities were eIF4E-independent. 

### 2.4. CABYV 3′-CITEs Vary in Sequence and Structure

The alignment in Figure 1B shows that the sequences of the two CABYV 3′-CITEs are different. Here, we analyzed their secondary structures in solution by Selective 2′-Hydroxyl Acylation analyzed by Primer Extension (SHAPE; [44]) using benzoyl cyanide (BzCN; [45]). This chemical quickly modifies flexible, possibly single-stranded nucleotides in a sequence-independent manner, forming 2′-O adducts that block reverse transcriptase. By incorporating the SHAPE reactivity data in the MC-fold server (http://www.major.iric.ca/MC-Fold), we obtained the mFold secondary structure predictions shown in Figure 5 (http://www.unafold.org/mfold/applications/rna-folding-form.php). The nucleotide variations found in the CXTEs of the other Asian isolates support the structure, as they are either in unpaired regions, or if they involve nucleotides proposed to be paired, they do not affect complementarity. The structure obtained for the CXTE of CABYV-Xin was similar to the one of the CXTE identified in MNSV-N [33], folding into two helices protruding from a central hub. Figure 5B shows the structure prediction obtained for the CMTE after analyzing the SHAPE experiments. The predicted secondary structure based on the SHAPE analysis showed that CMTE folded into two oppositely protruding helices with an additional short helix in the central part. The four nucleotides that were different in the CMTE of the recently sequenced isolate from Australia (Papua New Guinea) support the structure, since they only affect unpaired nucleotides.

### 2.5. Role of 3′-CITEs on Virus Multiplication Activity of CABYV 

To study the role in virus multiplication of the 3′-CITEs identified above, we modified two agroinfectious CABYV clones obtained from Spanish isolates, one unpublished cloned in pBIN61 and the other in pLJ89 [46], on the one hand by deleting its first 60 nt of the 3′-UTR (Δ-CITE), and, on the other hand, by exchanging its CMTE with the CXTE from CABYV-Xinjiang. The multiplication efficiencies of these clones were tested in melon cotyledons and *Nicotiana benthamiana* leaves. We analyzed the infiltrated leaves by Northern blot to be able to study virus multiplication through the detection of the subgenomic RNA (sgRNA). Figure 6A shows the viral RNAs (g/sgRNA) generated 3 days after agroinfiltration: in *N. benthamiana*, as well as in melon, the mutant virus without a 3′-CITE multiplied inefficiently, while the chimeric one containing the CXTE replicated with an efficiency higher than the wild-type virus (with its own CMTE). Similar results were obtained with both CABYV clones expressed from the two different binary plasmids. These results were in line with the ones from the in vivo translation experiments obtained with the luciferase constructs shown above (Figure 2). The size difference of 60 nt in the Δ-CITE clone can be observed in the sgRNA band that migrates slightly faster. 

The transcript that will start viral multiplication, synthesized after agrobacterium introduces its T-DNA into the cell, is capped [47]. Apart from this, poleroviruses have been described to have a viral protein linked to the 5′-end of their genome (VPg) that could be involved in virus protein synthesis [21,48,49] (Jiang and Laliberté, 2011; Reinbold et al., 2013; Van Der Wilk et al., 1997). Since 3′-CITEs control cap-independent translation, we wanted to study its importance in CABYV multiplication in the absence of 5′-cap or VPg. Thus, we analyzed the capacity of in vitro-transcribed capped and uncapped genomic RNA to multiply in melon protoplasts by quantifying the viral RNA by RT-qPCR. As can be observed in Figure 6B, the results obtained for virus multiplication when infecting protoplasts with capped or uncapped CABYV RNA were similar. Again, less CABYV RNA was detected in protoplasts electroporated with the mutant CABYV lacking the 3′-CITE. On the other hand, the chimeric virus with the CXTE showed a higher multiplication efficiency (1.6×) than the wild-type virus, congruently with the higher translation enhancement capacity observed for CXTE. We can conclude that CABYV multiplication (including translation) can occur in the absence of a 5′-cap or a VPg linked at the 5′-end of its genome, and additionally, that the 3′-CITEs identified in the CABYV 3′-UTR are required for efficient synthesis of the viral proteins and its multiplication. 

### 2.6. Possible 3′-CITE of New Brazilian CABYV Isolates

Recently, the complete sequences of three Brazilian isolates were published [10] and added to Genbank. These CABYV isolates were suggested to be recombinants with the CABYV-N isolate from France as the major parent (more than half of the 5′-genomic region), plus about 40% of its genome coming from a non-identified minor polerovirus parent (3′-genomic region including genes coding for CP and MP and 2/3 of the 3′-UTR). The end of the recombined region was mapped to the 3′-UTR, such that its last 59 nt again shared a high sequence identity (nearly 90%) with the French isolate [10]. On the other hand, the first 121 nt of the 3′-UTR had less than 50% identity with the French isolate. A phylogenetic analysis of all CABYV 3′-UTRs showed that the 3′-UTRs of these recombinant isolates form a separate group, independent of the Mediterranean or Asian isolates (Figure 7A). Separate alignments of the 3′-UTRs of Brazilian CABYVs with either Mediterranean or Asian isolates revealed that the first part of the Brazilian 3′-UTRs has no similarity with any of them (Appendix A). Thus, we wondered if Brazilian CABYVs had another type of 3′-CITE. 

To learn if the 5′-end of the 3′-UTR of Brazilian CABYV genome is indeed able to function as a 3′-CITE, we decided, based on 3′-UTR sequence alignments and structure predictions, to add the first 83 and 97 nucleotides of the 3′-UTR of CABYV-Brazil to the 5′-UTR-luc construct at its 3′-end. The cap-independent translation efficiencies of these constructs in melon protoplasts were measured. As observed in Figure 7B, presence of the shorter 83 nt fragment at the 3′-end of the luc gene did not result in an increase in translation as compared to the 5′-UTR-luc construct (first and last columns). On the other hand, the translation efficiency increased after the addition of the longer 97 nt fragment, although less efficiently (60%) than with the wild-type construct (CMTE-100%). Thus, we can conclude that the CABYV isolates newly identified in Brazil seem to have a 3′-CITE that is bigger and varies in sequence and structure (Figure 7C) from the ones of Mediterranean or Asian CABYV isolates. Again, the recombination hypothesis generated by the RDP4 software package predicted a recombination event with an unknown donor, including the first part of the 3′-UTR with very high statistical significance, strongly suggesting that the first part of the 3′-UTR of Brazilian CABYV including the 3′-CITE, was acquired through recombination (Appendix A). A nucleotide BLAST search with this acquired sequence did not result in any matches apart from Brazilian CABYV.

### 2.7. Searching for Possible 3′-CITEs in Polerovirus 3′-Ends

Since the polerovirus protein translation mechanisms are still unknown, we tried to identify possible 3′-CITEs in their genomic 3′ends. Based on comparisons of sequences and predicted structures, we identified CXTE-like structures in seven of them: melon aphid-borne yellows virus (MABYV), turnip yellows virus (TuYV), beet western yellows virus (BWYV), pepper vein yellows virus (PVYV), faba bean polerovirus (FBPV), beet chlorosis virus (BChV), and beet mild yellowing virus (BMYV) (Figure 8). Apart from structural similarity, the conservation of three pyrimidines (C/U) followed by three adenosines (A, only two for BYMV) in the loop of the second stem-loop structure was also observed. Thus, CXTE-like structures could be identified in 8 out of 20 polerovirus genomes in their 3′-ends, including CABYV, suggesting that polerovirus genome translation could generally depend on 3′-CITEs.

## 3. Discussion

In this report, we study CABYV protein synthesis in host cells. We started this study after the identification of a sequence inserted in the 3′-UTR of a new MNSV isolate (MNSV-N) able to break eIF4E-mediated resistance in melon that was highly similar to the first 60 nucleotides of the 3′-UTR of CABYV [33]. This sequence was shown to have the capacity to control cap-independent translation of MNSV in the absence of eIF4E. Here, we show that the first 60 nucleotides of the 3′-UTR of CABYV also function as an RNA element able to enhance cap-independent translation of its own genome. Surprisingly, the sequence of the first half of the 3′-UTR varies between CABYV isolates belonging to the different phylogenetic groups described. However, in spite of this, these sequences are functionally similar, having 3′-CITE activity. We show that these 3′-CITEs need the presence of the 5′-UTR *in cis* but do not need eIF4E for activity. Congruently, as these RNA elements enhance virus protein synthesis, they are required for efficient virus multiplication. We have not been able to identify a 5′–3′-interaction that is functionally important for 3′-CITE activity yet.

How polerovirus proteins are translated in the host cell is still unclear [50]. We propose that 3′-CITEs may be involved, as shown here for CABYV, since we identified, based on sequence/structure predictions, CXTE-like 3′-CITEs in 8 out of 20 polerovirus genomic 3′-ends (TuYV, MABYV, BChV, BWYV, PVYV, FBPV, and BMYV). For viruses of the genus *Luteovirus*, until recently belonging to the same family, 3′-CITEs were also identified, all similar to that of barley yellow dwarf virus (BYDV) translational enhancer (BTE) [28,51,52]. Although the BTE lies in an intergenic 3′- end region, not directly in the 3′-UTR, the sequences of several other 3′-CITEs have been shown to overlap with the coding region of the adjacent gene upstream of the 3′-UTR [32,53].

Here, we showed that uncapped CABYV-RNA is able to translate and multiply in melon protoplasts with a similar efficiency as capped CABYV-RNA, suggesting that the 5′-cap is not needed for initiation of infection from this input RNA. Polerovirus genomes are supposed to be uncapped [20]. Otherwise, its cap-independent, efficient translation and multiplication requires the 3′-CITEs identified in its 3′-UTRs. These results agree with the ones obtained with reporter constructs that showed that these sequences are able to enhance the translation efficiency. Since the CABYV mutant without a 3′-CITE is still able to multiply at low levels in melon protoplasts and *N. benthamiana* leaves, an additional mechanism for cap-independent translation of its proteins must exist.

VPgs from different viruses have been shown to interact with several eukaryotic translation initiation factors, suggesting a possible role in translation [49]. On the other hand, a role for potyviral VPgs in viral RNA stability and protection against being sent to the RNA silencing pathway has been proposed [54]. For some members of the polerovirus genus, a VPg has been proposed to be covalently linked to the genome at its viral 5′-end and to interact with different eIFs [21,49]. Early studies on potato leafroll virus (PLRV) showed evidence of the presence of a VPg linked to its genome [55]. Later, some amino acid sequence similarity between this VPg and the possible VPgs of another three poleroviruses, BMYV, BWYV, and CABYV, were described [48]. In spite of these reports, no direct studies on the existence of a VPg linked to the CABYV genome exists to date. Our finding that in vitro transcribed uncapped CABYV-RNA is able to multiply in protoplasts suggests that a VPg linked to the 5′-end is not essential for initiation of infection from this input RNA. However, the VPg may be required in the following steps of CABYV multiplication. 

Here, we identified different 3′-CITEs in each of the phylogenetically determined CABYV groups, Asian, Mediterranean, and Brazilian. Our recombination prediction analyses suggest that these 3′-CITEs have been acquired through recombination. The modularity and transferability of 3′-CITEs have been proposed before, as different types of 3′-CITEs have been found in a single genus, and the same type of 3′-CITE appeared in different virus genera [56]. Additionally, with our previous analysis on MNSV-N, we provided the first direct evidence that 3′-CITEs can be transferred between viral species through RNA recombination occurring in nature [33]. The recombination prediction analyses presented here suggest that the CABYV 3′-CITEs were also acquired through recombination, in this case with an unknown donor. Members of the polerovirus, luteovirus, and enamovirus genera, the former *Luteoviridae* family, are recombination prone, and they have been proposed as having emerged from intergeneric recombination events. Thus, several recombinants between poleroviruses and luteoviruses have been described, and CABYV itself has been proposed to derive from recombinations between polero- and enamo-like viruses [57,58,59]. Many CABYV recombinants have been identified, with the Brazilian isolates being the most recent ones [6,10,24,46].

3′-CITEs that differ in sequence and structure coincide in the general mechanistic steps involving recruitment of translation initiation factors at the 3′-CITE, and delivery of these near the translation start site through communication with the 5′-UTR [32]. Thus, when interchanged, 3′-CITEs can be active in very heterologous, but also in nearly identical viral genomes. To maintain an interchanged or newly acquired 3′-CITE in the viral genome, it must provide a selective advantage. Thus, with the acquisition of the CXTE from CABYV, the new MNSV isolate gained the ability to infect otherwise resistant melon [33]. Perhaps each of the three CABYV 3′-CITEs confers the virus a selective advantage in the geographical region where it evolved. For example, while comparing CXTE and CMTE translation enhancement activities, we found that CXTE works better in cucurbit cells, while CMTE does so in Arabidopsis cells. Thus, better multiplication in weeds may be the advantage for Mediterranean CABYV for maintaining CMTE, although this 3′-CITE is less efficient in cucurbits. Indeed, a recent study from our group indicated that weeds play an important role in CABYV diversification [23].

We can conclude that an RNA element located in the first set of nucleotides in the 3′-UTR of the CABYV polerovirus genome enhances its translation in host cells in the absence of a 5′-cap or VPg. Isolates from the three different phylogenetic CABYV groups, which evolved in different geographical regions, have different 3′-CITEs. Translation of other polerovirus genomes may also be 3′-CITE-dependent.

## 4. Materials and Methods

### 4.1. Luc-Constructs

The 20 nt long 5′-UTR was cloned after hybridization of two complementary primers including the T7 promoter followed by the 5′-UTR sequence and flanked by KpnI and NcoI restriction sites, after its digestion with these enzymes. The resulting fragment was directionally cloned into the KpnI/NcoI sites of the T7-luc plasmid (modified pGL3 [34], resulting in 5′-CABYV-luc). The 3′-UTR of CABYV-Sp was amplified by PCR with primers containing either XbaI or HpaI restriction sites from an unpublished full length Spanish CABYV isolate infecting melon and cloned in pTOPO. The 3′-UTR fragment was then directionally cloned into the XbaI/HpaI sites of the T7-luc or 5′-CABYV-luc plasmid [34], resulting in 3′-CABYV-Sp-luc or 5′-luc-3′-CABYV-Sp, respectively). The 3′-UTR of CABYV-Xin was obtained commercially (Genscript, Piscataway, NJ, USA) with restriction sites XbaI and HpaI, used for directional cloning into T7-luc or 5′-CABYV-luc plasmid (resulting in 3′-CABYV-Xin-luc or 5′-luc-3′-CABYV-Xin, respectively).

Deletion of the first 60 nt of both 3′-UTRs was achieved by amplification of the whole plasmid with complementary primers flanking this sequence on both sides but lacking this 60 nt. Subsequently, DpnI digestion was used to digest the input plasmid and select for the mutant plasmids (in vitro mutagenesis; [60]). 

The interchange of the first 60 nt of the 3′-UTR from CABYV-Sp for the ones of CABYV-Xin and *vice versa*, was also achieved by in vitro mutagenesis, using complementary primers including the sequence flanking this sequence on both sides and the 60 nt to be interchanged in between.

Analysis of shorter and longer 3′-CITE fragments in in vivo cap-independent translation assays delimited the optimal CABYV 3′-CITE lengths to 60 nt.

All new plasmid constructs were sequenced. These luc-inserts were amplified by PCR with the high-fidelity Prime Star HS DNA polymerase (Takara, Kusatsu, Japan) and transcribed in vitro (MEGAscript; Thermo Fisher Scientific, Waltham, MA, USA). Sequences of transcripts shown in Appendix A.

### 4.2. In Vivo Translation in Melon Protoplasts

Melon protoplasts were isolated from cotyledons as described in [34]. Brief description of in vivo translation: 5 μg of in vitro transcribed RNA (uncapped) was electroporated into 1 million protoplasts [43]. To minimize variations between samples, 2 μg of capped Renilla luciferase reporter RNA (pRL-null vector; Promega, Madison, WI, USA) were introduced along with the virus RNA. After 4–5 h incubation in the dark at 25 °C, protoplasts were precipitated and lysed with a final concentration of 0.5× PLB (Promega). Luciferase activities were measured with the Dual-GloTM Luciferase assay system (Promega) and Firefly activities normalized with respect to Renilla activities. These experiments were carried out at least five times for each construct.

### 4.3. RNA Structure Analysis

The protocol for the determination of the secondary structure of RNA fragments in solution by selective 2´-Hydroxyl acylation, analyzed by primer extension (SHAPE) was described in detail [61]. Briefly, the 3′-UTRs and 3′-CITEs (first 60 nt of 3′-UTRs) of CABYV-Sp and CABYV-Xin analyzed here, were inserted into the SHAPE cassette [62,63]). RNA, synthesized in vitro using MEGAshortscript TM Kit (Ambion, Austin, TX, USA), after treatment with 60mM of benzoyl cyanide (BzCN; Sigma-Aldrich, St. Louis, MI, USA) was reverse transcribed by primer extension of a radiolabeled primer. The cDNA fragments generated were resolved in an 8% denaturing polyacrylamide gel. Normalized BzCN reactivity values for each nucleotide position were calculated by SAFA Footprinting Software [64]. The RNA secondary structure was obtained using the MC-Fold computer program [65] after adding the SHAPE reactivity data. 

### 4.4. Construction and Multiplication Analysis of Mutant Viruses

The mutant viruses were constructed based on a full-length clone from an unpublished Spanish CABYV isolate infecting melon, cloned in pTOPO. The deletion of the first 60 nt of the 3′-UTR and the interchange of the CMTE in the 3′-UTR of CABYV-Sp for the CXTE from CABYV-Xin was performed by in vitro mutagenesis, using complementary primers including the sequence flanking both sides of the first 60 nt of the 3′-UTR, without this 60 nt, or with the CXTE 60 nt in between. Subsequently, DpnI digestion was used to digest the input plasmid and select for the mutant plasmids [61]. The obtained plasmids were sequenced. The agro-infectious mutant clones based on the binary vector pBIN61, were obtained by digestion of these CABYV-pTOPO mutants and CABYV-pBIN61 with HpaI followed by the exchange of the excised 2460 nt fragments of the 3′-end of the viral genomes (digestion at positions 3125 and 5585), resulting in CABYV-ΔCITE-pBIN61 and CABYV-CXTE-pBIN61. The agro-infectious mutant clones based on the binary vector pJL89 were constructed directly by in vitro mutagenesis performed on the agro-infectious CABYV-pJL89 plasmid CABYV-MEC12.1 [46]. All plasmids were sequenced (Appendix A).

CABYV wild-type and mutants were infiltrated into the abaxial side of expanded melon cotyledons or *N. benthamiana* leaves using the agro-infectious clones. The agro-infiltration and Northern blot protocols described in [66] were followed. TRI Reagent (Sigma-Aldrich) total RNA extractions of infiltrated leaves were performed after 3 days. For Northern blots, 2 μg (*N. benthamiana*) or 10 μg (melon) of total RNA were loaded on a denaturing gel. Viral genomic and subgenomic RNAs were detected by digoxigenin-labelled RNA probe complementary to a highly conserved CP-gene fragment [2]. 

For CABYV multiplication detection in melon protoplasts (see above), 5 μg of capped or uncapped RNA (transcribed in vitro, Appendix A) were electroporated into 1 million protoplasts, followed by 5 washes with 0.5 M Mannitol (pH 5.7) to get rid of RNA remaining outside the protoplasts. Negative controls were protoplasts incubated with RNA but without electroporation. Protoplasts were harvested after 24 h incubation at 25 °C with light. Total RNA was extracted with TRI reagent, purified by phenol-chloroform extraction, and followed by DNAseI treatment (Sigma-Aldrich). Quantification of viral RNA was performed as described by Rabadán et al. by one step RT-qPCR (NZYtech, Lisboa, Portugal) using 100 ng total RNA [46]. The results presented are the average of five independent experiments. 

## Figures and Tables

**Figure 1 ijms-23-12503-f001:**
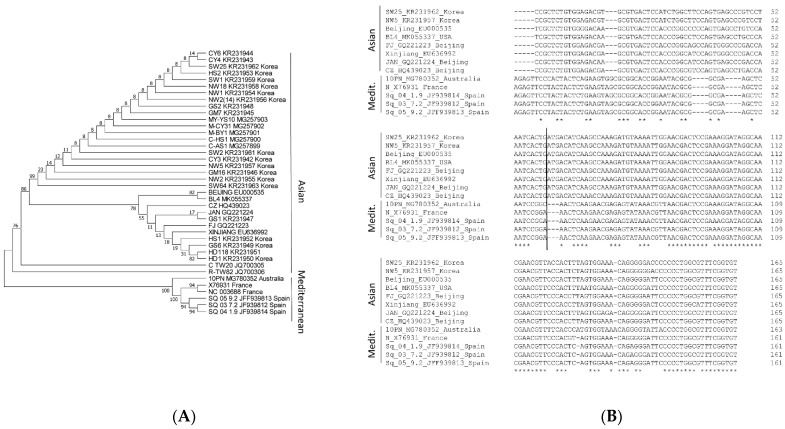
Evolutionary conservation study of the CABYV 3′-UTRs. (**A**) Phylogenetic relationship of 3′-UTR sequences of 40 CABYV isolates (Genbank accession number indicated for each isolate) and of the polerovirus *Chickpea chlorotic stunt virus* (CCSV) as the outgroup, inferred using the Maximum Likelihood method and Tamura-Nei model [36,37]. The tree with the highest log likelihood (−1029.01) is shown. The percentage of trees in which the associated taxa clustered together is shown next to the branches. Initial tree(s) for the heuristic search were obtained automatically by applying Neighbor-Join and BioNJ algorithms to a matrix of pairwise distances estimated using the Maximum Composite Likelihood (MCL) approach, and then selecting the topology with superior log likelihood value. This analysis involved 41 nucleotide sequences. There was a total of 186 positions in the final dataset. Evolutionary analyses were conducted in MEGA X [38]. (**B**) Clustal Omega (https://www.ebi.ac.uk/Tools/msa/clustalo/) sequence alignment of the 3′-UTRs of 8 CABYV isolates belonging to the Asian group and all of the ones available in Genbank that belong to the Mediterranean group. Asterisks mark invariable nucleotides. The vertical line marks the 60th nucleotide.

**Figure 2 ijms-23-12503-f002:**
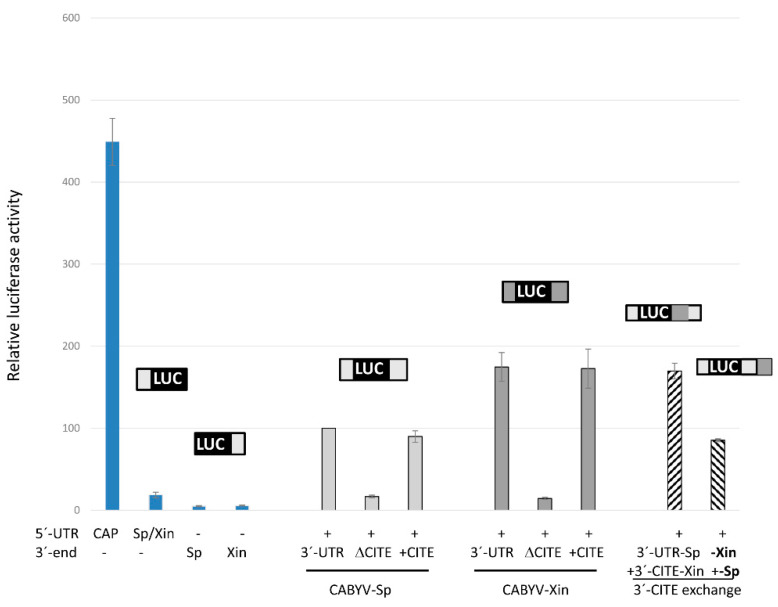
The 3′-UTRs of the CABYV genome contain 3′-CITEs. In vivo cap-independent translation efficiency of different luc-constructs assayed in melon protoplasts. Vertical columns represent measured luciferase activity (corresponding to the translation efficiency) relative to the activity obtained with the construct 5′-luc-3′-UTR of CABYV-Sp (100%). The first column shows the translation efficiency of a capped construct (with plasmid sequence flanking the 5′- and 3′-UTRs). Above the columns, a schematic drawing of the respective constructs assayed is shown (solid grey for CABYV-Sp/Xin sequences flanking luc, lined for chimeric 3′-UTRs). In the penultimate column, the 3′-CITE of CABYV-Sp in its 3′-UTR was interchanged for the 3′-CITE of CABYV-Xin, while in the last column, the 3′-CITE of CABYV-Xin in its 3′-UTR was interchanged for the 3′-CITE of CABYV-Sp. Error bars are +/−SD.

**Figure 3 ijms-23-12503-f003:**
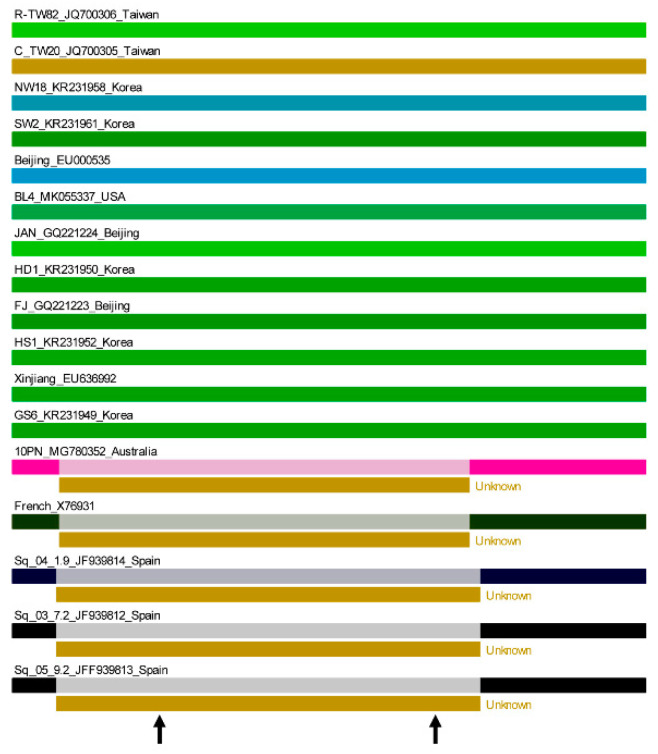
Recombination prediction for the 3′-ends of the CABYV genome. Recombination hypothesis generated by the RDP4 software package (http://web.cbio.uct.ac.za/~darren/rdp.html). This computer program characterizes recombination events in sequence alignments using several different recombination analysis methods and tests for recombination hot-spots. The sequences included in this analysis performed using RDP, GENECONV, Maximum Chi-square (MaxChi), BootScan, SisterScan (SiScan), Chimaera, and 3Seq, include the last 50 nt of ORF5 plus the following 3′-UTRs of different CABYV isolates belonging to the Mediterranean and Asian groups (Genbank accession number indicated for each isolate). Default settings with a Bonferroni corrected *p*-value cut-off of 0.01 were applied. To reduce the possibility of false detection of recombination, only recombination events supported by at least two methods were selected. RDP4 colors similar sequences with similar colors. The arrow on the left marks the position of the start of the 3′-UTR sequence, and the one on the right, the approximate end of the variable region, from which the UTR sequences of Asian and Mediterranean isolates start to be conserved. The statistical significance was very high, with a *p*-value < 0.00001.

**Figure 4 ijms-23-12503-f004:**
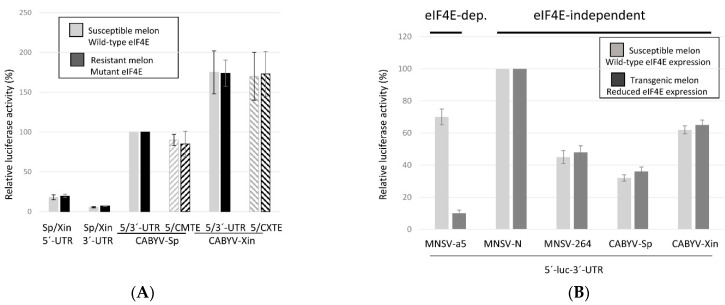
eIF4E-dependence of CABYV 3′-CITEs activities. In vivo cap-independent translation efficiencies of different luc-constructs assayed in melon protoplasts. Below the columns, the respective constructs assayed are explained. (**A**) Protoplasts were prepared from wild-type susceptible and resistant melon (containing a single amino acid change in eIF4E). Vertical columns represent measured luciferase activity relative to the activity obtained with the construct 5′-luc-3′-UTR of CABYV-Sp (100%) (in light grey for susceptible and dark grey for resistant melon). The columns showing the efficiencies obtained with the constructs 5′-UTR-luc-3′-CITE are lined. (**B**) Protoplasts were prepared from wild-type and transgenic melon eIF4E silenced for eIF4E (light grey and dark grey columns, respectively). Vertical columns represent measured luciferase activity relative to the activity obtained with the construct 5′-luc-3′-UTR of MNSV-N (100%), which just as the one of MNSV-264, serves as the eIF4E-independent control, while that of MNSV-Mα5 is the eIF4E-dependent control. Error bars are +/−SD.

**Figure 5 ijms-23-12503-f005:**
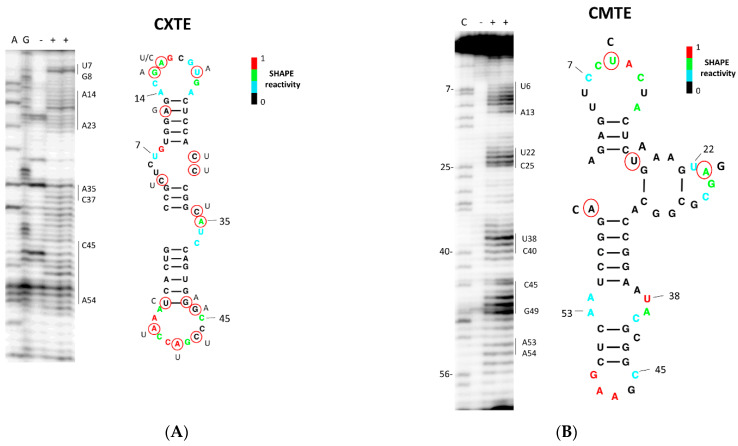
Secondary structure probing of the CABYV 3′-CITEs. Structure probing by SHAPE of the first 60 nt of the 3′-UTR corresponding to the 3′-CITEs of CABYV. (**A**) CXTE structure of CABYV-Xin, representing the Asian isolates. (**B**) CMTE structure of CABYV-Sp, representing the Mediterranean isolates. The secondary structures of the 3′-CITEs were obtained by superimposing the SHAPE reactivity of nucleotides on the secondary structure predicted by MCfold/mFold. Color-coded bases indicate the levels of BzCN modification, with warmer colors indicating greater modification (inset). The nucleotide variations found with respect to the 3′-CITEs of the other Asian (**A**) or Mediterranean (**B**) CABYV isolates are shown (red circle). All of them support the proposed secondary structure. At the left of each structure, the denaturing PAGE of the separated primer extension products of RNA treated (last lane (+)) or untreated (third lane, (−)) with benzoyl cyanide (BzCN) is shown. The sequencing ladders were generated by reverse transcription of unmodified RNA in the presence of dideoxyCTP (ddCTP; lane G), ddTTP (lane A), or ddGTP (lane C). The positions in the PAGE corresponding to the loops of the two stem-loop structures are marked on the right side.

**Figure 6 ijms-23-12503-f006:**
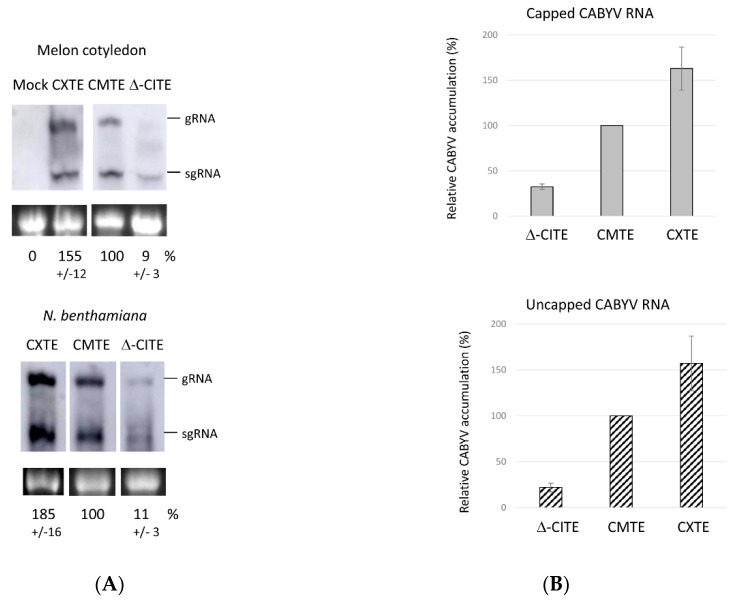
Role of 3′-CITEs in virus multiplication. (**A**) Northern blot of total RNA extracts from melon cotyledon (upper blot) and *N. benthamiana* leaves (lower blot) three days after agroinfiltration with CABYV-pBIN61 clones. The genomic RNA (gRNA, 5.7 kb) and subgenomic RNA (sgRNA, 2.5 kb) were detected using an RNA probe complementary to a CP-gene fragment. The lower panel shows stained ribosomal RNA as a loading control. Below these panels the average percentage (% +/−SD) of multiplication calculated from five independent experiments are indicated. (**B**) Quantification of CABYV accumulation in melon protoplasts relative to the wild-type virus (CMTE = 100%) measured by RT-qPCR. The upper panel shows virus multiplication obtained after infecting protoplasts with capped CABYV-RNA, the lower panel with uncapped CABYV-RNA. The background CABYV-RNA quantified in the negative control (protoplasts incubated with RNA but without electroporation) was subtracted from the values obtained for the electroporated samples before calculating the percentages. The results presented are the average of five independent experiments.

**Figure 7 ijms-23-12503-f007:**
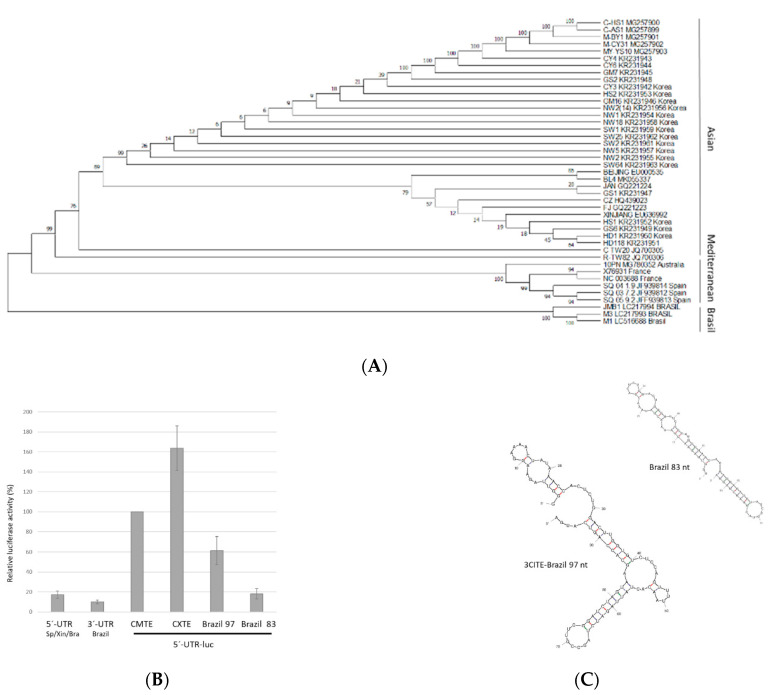
Brazilian CABYV isolates also have a 3′-CITE. (**A**) Phylogenetic relationship of 3′-UTR sequences of CABYV isolates (43) available in Genbank (accession number indicated for each isolate) and of the polerovirus *Chickpea chlorotic stunt virus* (CCSV) as the outgroup inferred using the Maximum Likelihood method and Tamura-Nei model [36,37]. The tree with the highest log likelihood (−1271.12) is shown. The percentage of trees in which the associated taxa clustered together is shown next to the branches. Initial tree(s) for the heuristic search were obtained automatically by applying Neighbor-Join and BioNJ algorithms to a matrix of pairwise distances estimated using the Maximum Composite Likelihood (MCL) approach, and then selecting the topology with superior log likelihood value. This analysis involved 44 nucleotide sequences. There were a total of 188 positions in the final dataset. Evolutionary analyses were conducted in MEGA X [38]. (**B**) In vivo cap-independent translation efficiency of different luc-constructs assayed in melon protoplasts. Vertical columns represent measured luciferase activity (corresponding to the translation efficiency) relative to the activity obtained with the construct 5′-luc-3′-CITE of CABYV-Sp (100%). Below the columns, the respective constructs assayed are explained. Error bars are +/−SD. (**C**) Secondary structure predictions for the first 83 and 97 nt of the 3′-UTR invariant in the three Brazilian isolates using mFold.

**Figure 8 ijms-23-12503-f008:**
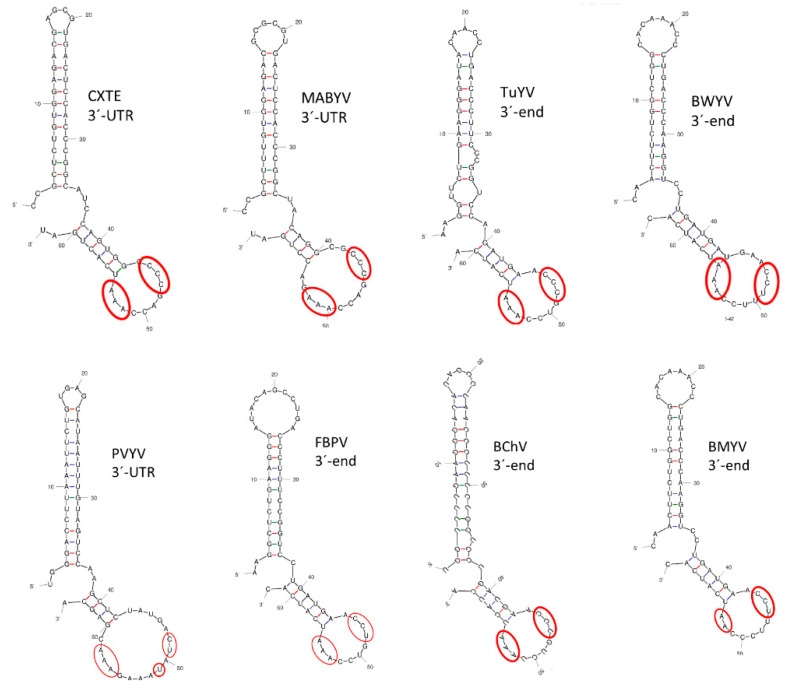
Secondary structure predictions of possible polerovirus 3′-CITEs. Structural similarities with the Asian-CXTE (predictions by mFold) of genomic 3′-ends of several poleroviruses: melon aphid-borne yellows virus (MABYV, 3′-UTR), turnip yellows virus (TuYV, 3′-end-10 nt of ORF5), beet western yellows virus (BWYV, 3′-end-25 nt of ORF5), pepper vein yellows virus (PVYV, 3′-UTR), faba bean polerovirus (FBPV, 3′-end-10 nt of ORF5), beet chlorosis virus (BChV, 3′-end-24 nt of ORF5), beet mild yellowing virus (BMYV, 3′-end-39 nt of ORF5). Conserved nucleotides in the loop of the second stem-loop structure are marked with a red circle: 3 pyrimidines (C/U) and 3 adenosines (A, only two for BYMV).

## Data Availability

Not applicable.

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
