# Peer review of "Different RNA Elements Control Viral Protein Synthesis in Polerovirus Isolates Evolved in Separate Geographical Regions"

_ijms, 2022, doi:10.3390/ijms232012503_

Round 1
Reviewer 1 Report
In this study, the Authors investigate a putative 3’CITE in CABYV polerovirus and determined that it contributes to efficient translation of viral proteins and is important for virus accumulation in plants. The work is an extension of a previous publication showing that a 3’CITE in MNSV (Tombusviridae) was likely derived from poleroviruses. The results of the current study are novel and significant, because they represent the first data indicating that an RNA element in the 3’UTR of a polerovirus can contribute to viral protein synthesis. Additionally, this 3’CITE appears to also be present in other polerovirus members. Overall, the results are largely solid and the conclusions reasonable. However, there are areas that should be improved prior to acceptance. Suggestions for improvement are provided below.
COMMENTS:
Title: Unhyphenate polerovirus – it is the focus of the study and should be shown as a single word
Abstract: “Importantly, efficient virus multiplication depended on the 3´-CITE activity and occurred in the absence of a 5´-cap or VPg linked to its genomic 5´-end.” This is misleading, as it suggests that VPg is not required for efficient virus replication. The VPg is likely added to nascent genomes synthesized after the initial infection with in vitro transcripts, and as in other VPg-containing viruses, it may be essential to priming viral RNA synthesis. You would need to inactivate the VPg in the genome in order to claim that it is not required. Same problem with a similar statement in Discussion Ln443: “Here we showed that uncapped CABYV-RNA is able to translate and multiply in melon protoplasts with similar efficiency as capped CABYV-RNA, suggesting that the 5´-cap is not needed for virus multiplication.” And Discussion ln460: “Our finding that in vitro transcribed uncapped CABYV-RNA is able to multiply in protoplasts suggests that a VPg linked to the 5´-end is not essential for CABYV multiplication, although we cannot exclude that the presence of VPg could make translation and multiplication more successful. “ Again, the results show only that “input inoculum genomes” does not need a VPg to initiate an infection, not that the VPg is not needed for replication. All of these statements need to be corrected.
Abstract: “In CABYV isolates belonging to the three phylogenetic groups identified so far, the 3´-CITEs differ supporting their modularity and transferability.” – not sure the differences support “transferability” – because the variations observed likely occurred by divergent evolution of an initial virus containing this 3’-CITE (and not transfer/recombination).
Ln26 – Define cucurbit aphid-borne yellows virus (CABYV) when first used in the Introduction section
Ln 27 – change to “virus”
Fig2. Add and test capped versions of negative controls to determine how the cap-independent levels compare to actual cap-dependent levels. Currently, it is not known if the observed levels for cap-independent translation are comparable to cap-dependent levels.
Fig6A – repeats and statistical analysis (+/-SD) need to be applied to the quantification
Fig7A – resolution needs to be improved – base identity are difficult to read
Discussion: ln431 – Please speculate as to why the 5’UTR is needed – are there any possible 5’-3’ RNA-RNA interactions involved as for other 3’CITEs?
Ln 463L “Interestingly, BYDV, known to have a 3´-CITE controlling its protein translation, was shown to have a VPg linked to its genomic 5´-end [56].” This statement is not correct. A later study by Allen Miller’s lab showed that BYDV is neither 5’capped nor has a VPg. DOI: 10.1006/viro.1998.9507 Therefore, this sentence and the next sentence need to be corrected/removed.
Discussion – It should be mentioned/discussed that the delta-CITE CABYV mutant can still accumulate at low levels in plants and protoplasts, therefore there may be other translation elements that exist and allow for the low levels of accumulation.
General English and formatting: needs very heavy editing for correct use of English. Also, text formatting is not consistent, see changes in font size and line spacing throughout. Careful English and formatting editing needs to be done on entire manuscript
Author Response
Response referee 1
Thank you very much for your revision, it has greatly improved the manuscript. Please, find enclosed a revised version of the manuscript according the referee's comments. We have tried to incorporate all proposed suggestions and answer every point raised by the referees:
Title: Unhyphenate polerovirus – it is the focus of the study and should be shown as a single word
The formatting of the title was been changed by the journal. We will ask them to keep “polerovirus” as a single word when editing the manuscript
Abstract: “Importantly, efficient virus multiplication depended on the 3´-CITE activity and occurred in the absence of a 5´-cap or VPg linked to its genomic 5´-end.” This is misleading, as it suggests that VPg is not required for efficient virus replication. The VPg is likely added to nascent genomes synthesized after the initial infection with in vitro transcripts, and as in other VPg-containing viruses, it may be essential to priming viral RNA synthesis. You would need to inactivate the VPg in the genome in order to claim that it is not required.
Thank you for this important observation. To avoid misleading conclusions, the sentence was shortened to: “Efficient virus multiplication depended on 3´-CITE activity.“
Same problem with a similar statement in Discussion Ln443: “Here we showed that uncapped CABYV-RNA is able to translate and multiply in melon protoplasts with similar efficiency as capped CABYV-RNA, suggesting that the 5´-cap is not needed for virus multiplication.”
Second part of sentence was changed to: “suggesting that the 5´-cap is not needed for initiation of infection from this input RNA.” The next sentence “This experiment supports that indeed the CABYV genome may be uncapped, since the 5´-cap is not required for virus multiplication” was deleted.
And Discussion ln460: “Our finding that in vitro transcribed uncapped CABYV-RNA is able to multiply in protoplasts suggests that a VPg linked to the 5´-end is not essential for CABYV multiplication, although we cannot exclude that the presence of VPg could make translation and multiplication more successful. “ Again, the results show only that “input inoculum genomes” does not need a VPg to initiate an infection, not that the VPg is not needed for replication. All of these statements need to be corrected.
Thank you again for this important observation. Changed to: “Our finding that in vitro transcribed uncapped CABYV-RNA is able to multiply in protoplasts suggests that a VPg linked to the 5´-end is not essential for initiation of infection from this input RNA. However, the VPg may be required in the following steps of CABYV multiplication."
Abstract: “In CABYV isolates belonging to the three phylogenetic groups identified so far, the 3´-CITEs differ supporting their modularity and transferability.” – not sure the differences support “transferability” – because the variations observed likely occurred by divergent evolution of an initial virus containing this 3’-CITE (and not transfer/recombination).
The recombination prediction with the RDP4 software package, which implements several recombination-detecting algorithms, detects phylogenetic incongruences in the different CABYV 3´-CITEs and suggests they were transferred by recombination with an unknown donor. There is nearly no homology between the first 60-90 nts of the 3´-UTR of the isolates belonging to the three different phylogenetic groups, while the rest of the 3´-UTR shows high homology (Fig. 1B and suppl. Fig. 5). This should tilt the balance from divergent evolution to transfer by recombination. The sentence has been changed to: “In CABYV isolates belonging to the three phylogenetic groups identified so far, the 3´-CITEs differ, and recombination prediction analyses suggest that these 3´-CITEs have been acquired through recombination with an unknown donor.”
Ln26 – Define cucurbit aphid-borne yellows virus (CABYV) when first used in the Introduction section done
Ln 27 – change to “virus” done
Fig2. Add and test capped versions of negative controls to determine how the cap-independent levels compare to actual cap-dependent levels. Currently, it is not known if the observed levels for cap-independent translation are comparable to cap-dependent levels.
Done. Text added: Its translation efficiency was 2.5-3.5-times lower than that of a capped construct. In Legend: The first column shows the translation efficiency of a capped construct (with plasmid sequence flanking the 5´- and 3´-UTRs).
Fig6A – repeats and statistical analysis (+/-SD) need to be applied to the quantification
ADDED… the average percentage (% +/- SD) of multiplication calculated from five independent experiments are indicated.
Fig7A – resolution needs to be improved – base identity are difficult to read
Discussion: ln431 – Please speculate as to why the 5’UTR is needed – are there any possible 5’-3’ RNA-RNA interactions involved as for other 3’CITEs?
We identified some 5´-3´-end complementary sequence stretches (also looking in ORF1, since 5´-UTR is only 20 nt long), but these were not involved in 3´-CITE mediated translation control. The following sentence was added: “We have not been able to identify a 5´-3´-interaction that is functionally important for 3´-CITE activity yet.”
Ln 463L “Interestingly, BYDV, known to have a 3´-CITE controlling its protein translation, was shown to have a VPg linked to its genomic 5´-end [56].” This statement is not correct. A later study by Allen Miller’s lab showed that BYDV is neither 5’capped nor has a VPg. DOI: 10.1006/viro.1998.9507 Therefore, this sentence and the next sentence need to be corrected/removed.
Thank you again for this important observation. “Interestingly, BYDV, known to have a 3´-CITE controlling its protein translation, was shown to have a VPg linked to its genomic 5´-end [56]. Thus, translation control by a 3´-CITE does not seem to exclude the presence of a VPg at the opposite side of the viral genome, even when 3´-CITE activity requires the presence of the 5´-UTR in cis, as is the case for CABYV and BYDV” has been removed.
Discussion – It should be mentioned/discussed that the delta-CITE CABYV mutant can still accumulate at low levels in plants and protoplasts, therefore there may be other translation elements that exist and allow for the low levels of accumulation.
Yes, this is what we think, and therefore we always talk about the “translation enhancement” activity of the 3´-CITE. The following sentence has been added: “Since the CABYV mutant without a 3´-CITE is still able to multiply at low levels in melon protoplasts and N. benthamiana leaves, an additional mechanism for cap-independent translation of its proteins must exist.”
General English and formatting: needs very heavy editing for correct use of English. Also, text formatting is not consistent, see changes in font size and line spacing throughout. Careful English and formatting editing needs to be done on entire manuscript
The formatting has been changed by the journal. We will ask them to be careful.
The use of the English language has been reviewed by a native speaker.

Reviewer 2 Report
Miras et al performed phylogenetic analyses on the 3'UTR sequences from different CABYV isolates, and showed that the first half of the 3'UTR is highly variable. Previous reports claimed that the first half of the 3'UTR of related viral mRNAs might promote cap-independent translation of the viral genes, and here, authors tried to show that the same holds for the CABYV 3'UTRs.
Major:
This study relied on luciferase assay to quantify the "translation efficiency" of varying uncapped mRNAs that were in vitro transcribed and electroporated in to cells. From the described methods by the authors (Line 530-539), authors did not control for mRNA levels in the sample, e.g. using quantitative RT-PCR. Normalizing the luciferase activity to the mRNA levels is the key to rule out any mRNA stability variations caused by the 3'UTR elements. Without such analysis, whether the observed effect was due to differences in translation or mRNA stability was not clear, thus authors' interpretation of the data were not valid.
Fig. 2, a capped mRNA control is missing, which is key to understand the exact efficiencies of the cap-independent translation promoted by the putative 3' elements. Again, the Luc activity should be normalized by mRNA level.
Minor:
Some of the figures are too small to be readable, e.g. Fig 7 and 8.
Line 96, change "these" to "that".
Line 149, define CXTE.
Line 527-529, authors should provide the plasmid/mRNA sequences for all the assayed constructs.
Author Response
Response referee 2
Miras et al performed phylogenetic analyses on the 3'UTR sequences from different CABYV isolates, and showed that the first half of the 3'UTR is highly variable. Previous reports claimed that the first half of the 3'UTR of related viral mRNAs might promote cap-independent translation of the viral genes, and here, authors tried to show that the same holds for the CABYV 3'UTRs.
Thank you very much for your revision, it has greatly improved the manuscript. Please, find enclosed a reviewed version of the manuscript according the referee's comments. We have tried to incorporate all proposed suggestions and respond to every point raised by the referees:
Major:
This study relied on luciferase assay to quantify the "translation efficiency" of varying uncapped mRNAs that were in vitro transcribed and electroporated in to cells. From the described methods by the authors (Line 530-539), authors did not control for mRNA levels in the sample, e.g. using quantitative RT-PCR. Normalizing the luciferase activity to the mRNA levels is the key to rule out any mRNA stability variations caused by the 3'UTR elements. Without such analysis, whether the observed effect was due to differences in translation or mRNA stability was not clear, thus authors' interpretation of the data were not valid.
We think the analysis of RNA stability in our experimental conditions gives no reliable results (we have tried to do this experiment before), since we perform in vivo translation assays in protoplasts (and not in vitro assays in wheat germ extract): The protoplasts are electroporated with 5 mg of RNA and even after more than five wash steps, we still find RNA outside the protoplasts (that has not entered the protoplast). Thus, in RT-qPCR analyses the RNA remaining outside the protoplasts covers the low amount of RNA that enters the protoplasts and no conclusions about mRNA stability can be drawn.
Apart from this, we don’t think that our results correspond to RNA stability variations: In the in vivo luciferase translation experiments, constructs with complete 3´-UTR flanking the 5´-luc, and those with only the first 60 nt of the 3´-UTR, gave similar efficiencies, while deleting the 60 nt of the 3´-UTR reduced translation. Additionally, the translation efficiencies obtained with the chimeric 3´-UTRs are congruent with the CXTE being more efficient in melon, but not in Arabidopsis. Finally, the results obtained with the reporter constructs are in line with the results obtained for virus multiplication. For all of these reasons, we don’t think that these results correspond to RNA stability variations.
Additionally, many publications analyzing 3´-CITEs using reporter constructs do not study the mRNA stability, for example:
Kraft, Peterson, Cho , Wang , Hui Rakotondrafara , Treder 9, and Miller. 2019. Pathogens, 8, 28; DOI:10.3390/pathogens8010028
Gao and Simon, 2017. Virology 510:194–204, DOI: 10.1016/j.virol.2017.07.021
Liu and Simon. 2022. Journal of Virology 96 ,7 DOI: 10.1128/jvi.01736-21
Sheets and Redinbaugh, 2006. Virology 350: 171–183DOI: 10.1016/j.virol.2006.02.004
Stupina, Meskauskas, McCormack, Simon. 2008. RNA 2008 14: 2379-2393, DOI: 10.1261/rna.1227808
Wang, Treder and Miller, 2009. Journal of Biological Chemistry Vol. 284, NO. 21, pp. 14189–14202, May 22, DOI: 10.1074/jbc.M808841200
Fig. 2, a capped mRNA control is missing, which is key to understand the exact efficiencies of the cap-independent translation promoted by the putative 3' elements. Again, the Luc activity should be normalized by mRNA level.
Done. Text added: Its translation efficiency was 2.5-3.5-times lower than that of a capped construct. In Legend: The first column shows the translation efficiency of a capped construct (with plasmid sequence flanking the 5´- and 3´-UTRs).
Minor:
Some of the figures are too small to be readable, e.g. Fig 7 and 8.
The quality of figures has been improved
Line 96, change "these" to "that".
We think that “these” is correct. A coma was added and “previously” deleted: “Following phylogenetic analyses of the complete genome of CABYV isolates, these were classified into two groups, Asian and Mediterranean”
Line 149, define CXTE.
Done. “Thus, these results confirm that the 3´-CITE identified as active RNA element in MNSV-N (named CXTE, X for Xinjiang) is also working as a 3´-CITE in Asian CABYV isolates, while Mediterranean isolates also have a 3´-CITE that differs in sequence, now named CMTE (M for Mediterranean).”
Line 527-529, authors should provide the plasmid/mRNA sequences for all the assayed constructs.
A table with the sequences (Suppl. Table 1) has been added.

Round 2
Reviewer 1 Report
The authors have adequately addressed the comments of this reviewer and the manuscript is now acceptable for publication.
Author Response
Referee 1:
The authors have adequately addressed the comments of this reviewer and the manuscript is now acceptable for publication.
Again thank you very much for your revision, the manuscript has greatly improved with your comments.
Reviewer 2 Report
Despite the multiple reasons from the authors to not normalize the Luc activity by mRNA levels, I still feel this is a compromised solution. After all, authors are not certain about if the same amount of the varying mRNAs were transfected into the cells, nor if the mRNAs were degraded with different rates. Therefore, I still have reasons to question the measured translation activities, thus the 3'-CITE activities, and the conclusions in the paper.
Author Response
Referee 2:
Despite the multiple reasons from the authors to not normalize the Luc activity by mRNA levels, I still feel this is a compromised solution. After all, authors are not certain about if the same amount of the varying mRNAs were transfected into the cells, nor if the mRNAs were degraded with different rates. Therefore, I still have reasons to question the measured translation activities, thus the 3'-CITE activities, and the conclusions in the paper.
We don’t think that our results correspond to RNA stability variations: The 3´-CITE of CABYV-Xin, CXTE, was reported to give the resistance-breaking isolate MNSV-N the capacity to overcome the eIF4E-mediated resistance in melon (Miras et al., New Phytologist, 2014, doi:10.1111/nph.12650). The resistance-breaking capacity of MNSV-N was clearly localized in the 60 nts newly inserted in its 3´-UTR, a sequence coming from a 3´-UTR of Asian CABYV. This sequence was demonstrated to act as a 3´-CITE allowing MNSV-N protein synthesis in the absence of eIF4E. We think that differences in RNA stability would not allow an isolate to overcome a eIF4E-mediated resistance. In the present manuscript we analyzed, if the sequence that had been acquired by MNSV-N functioning as 3´-CITE, was also active in CABYV. The results suggest that it does so. Another result against the argument that the effects of the CABYV 3´-CITEs on translation are the result of RNA stability differences, is the finding that the translation enhancement activity of CXTE was higher than that of CMTE in melon, but lower than CMTE in Arabidopsis protoplasts. Also, the results obtained with the reporter constructs are in line with the results obtained for virus multiplication. For all of these reasons, we don’t think that these results correspond to RNA stability variations.
Apart, experimental problems exist: We always perform our translation experiments in vivo in protoplasts of a viral host, because we believe these conditions are better adjusted to the natural conditions of viral multiplication in the host cell. The protoplasts are always electroporated with 5 mg of in vitro transcribed RNA, independent of the construct. In each protoplasts preparation experiment the 5´-luc-3´-UTR constructs of CABYV-Sp and CABYV-Xin are also analyzed as positive controls. Thus, the amount of RNA entering the protoplasts of each preparation should be the same for all constructs. Additionally, to minimize variations between samples, 2 mg of capped Renilla luciferase reporter RNA were introduced along with the RNA of each construct. We tried to perform RNA stability analyses in our experimental conditions, this is in vivo translation in protoplasts, but we obtained no reliable results. The RNA remaining outside the protoplasts was always higher than the low amount of RNA that entered the protoplasts and no conclusions about transfected RNA and its stability could be drawn.
Round 3
Reviewer 2 Report
I don't have further comments on this point, and will leave it to the editorial team to decide on the next steps.